# Risk of Falling in a Timed Up and Go Test Using an UWB Radar and an Instrumented Insole

**DOI:** 10.3390/s21030722

**Published:** 2021-01-21

**Authors:** Johannes C. Ayena, Lydia Chioukh, Martin J.-D. Otis, Dominic Deslandes

**Affiliations:** 1Communications and Microelectronic Integration Laboratory (LACIME), Department of Electrical Engineering, École de Technologie Supérieure, 1100 Rue Notre-Dame Ouest, Montréal, QC H3C 1K3, Canada; cossoun-johannes.ayena.1@ens.etsmtl.ca (J.C.A.); Dominic.Deslandes@etsmtl.ca (D.D.); 2Laboratory of Automation and Robotic Interaction (LAR.i), Department of Applied Science, University of Quebec at Chicoutimi, 555 Blvd of University, Chicoutimi, QC G7H 2B1, Canada; martin_otis@uqac.ca

**Keywords:** fall detection, biomedical monitoring, TUG, UWB radar, gait parameters, non-contact

## Abstract

Previously, studies reported that falls analysis is possible in the elderly, when using wearable sensors. However, these devices cannot be worn daily, as they need to be removed and recharged from time-to-time due to their energy consumption, data transfer, attachment to the body, etc. This study proposes to introduce a radar sensor, an unobtrusive technology, for risk of falling analysis and combine its performance with an instrumented insole. We evaluated our methods on datasets acquired during a Timed Up and Go (TUG) test where a stride length (SL) was computed by the insole using three approaches. Only the SL from the third approach was not statistically significant (*p* = 0.2083 > 0.05) compared to the one provided by the radar, revealing the importance of a sensor location on human body. While reducing the number of force sensors (FSR), the risk scores using an insole containing three FSRs and y-axis of acceleration were not significantly different (*p* > 0.05) compared to the combination of a single radar and two FSRs. We concluded that contactless TUG testing is feasible, and by supplementing the instrumented insole to the radar, more precise information could be available for the professionals to make accurate decision.

## 1. Introduction

Considerable efforts are being made to expand our knowledge and capacities to solve biomedical problems. Nowadays, some studies emphasize the need to successfully integrate engineering into life and health sciences [1]. Rising healthcare costs and an aging population are factors influencing research in the medical sector, mainly for the development of systems for assisted living and the creation of smart homes [2]. Ultimately, these systems must be able to fulfill several functions: detection of the position of the sick person, early forecasting of health problems in the elderly, measurement of vital signs, assistance, and control in real time, generation of instant alerts for health workers or families, etc., and the monitoring of specific events and activities. Specifically, the aging population is growing rapidly in developed countries, and suffers from high death rates due to age-related disorders and falls [3]. Researchers are therefore challenged to find new efficient fall detection and monitoring solutions to minimize injuries, accidents, and the cause of chronic disability, not to mention the psychological consequences [4]. Detecting falls at the time of the incident is crucial among people with gait and balance problems. Continuous and autonomous remote monitoring of activities is necessary and may increase the safety and reduce the risk of mortality in elderly people alone at home [5,6,7]. For doing so, many works have been devoted to the use of different technologies to detect falls [8]. Several major classes of devices are proposed for biomechanical control, such as portable contact systems, non-portable ambient devices, and vision-based devices.

Fall detection systems with contact using inertial sensors have been investigated in the literature [9], especially portable sensors integrated to detect the movement of the patient, such as accelerometer [10], gyroscope sensors [11], acoustic sensors [12], and the use of smartphone applications offered [13,14]. However, their suggested technologies have drawbacks and constraints for the users. They are not often worn daily by the elderly, as they need to be recharged and removed from time-to-time. Moreover, they are very sensitive to noise and interference, and the detection and accuracy are often affected by their position and attachment on the human body. Other researchers favor the use of instrumented insoles [15] containing pressure sensors since they are discrete, inexpensive and sensitive. However, they have to face several challenges such as the biocompatibility of the materials used in the manufacturing process, the fixation of components, energy consumption and transmission, data transfer, the mobility and stability of the insole in the shoe with a long lifespan. Thus, it becomes necessary to think of other alternatives systems for also facilitating the locomotion of the individual during the assessment of a balance problem.

As a result, systems based on passive infrared light (PIR) [16,17] or Wi-Fi [18,19,20] are finding new perspectives in the falls assessment domain. Most of them do not present home privacy issues; however, the monitoring is often affected by clothing, and often requires the whole body to be in the field of vision. Moreover, the localization of finite points is generally insufficient to capture the human movements with more detail. Other systems based on vision using the capture of videos and images (depth cameras and RGB cameras) are exploited for the detection and localization of falls [21,22,23]. These systems are focused on the detection of the position and posture of the patient after a complex signal processing. These methods therefore require numerous cameras, calibration, and preprocessing techniques for the recognition of posture and the construction of images.

Currently, in the medical sphere, other technologies are exploited, in which radiofrequencies are mainly used. Among them, we note the continuous wave (CW) and ultra-wideband (UWB) radars. Indeed, microwaves are used for the communication of medical devices, whether external or implanted. The combination of communication standards and reconfiguration of systems have led to an evolution of radar architectures and the development of new technological concepts that can reduce the health problems. The advantage of radars for fall monitoring [22,24,25] is that these systems offer confidentiality and autonomy since the detection of a fall does not require the wearing of instruments [26,27]. Furthermore, radar systems are energy efficient systems with low consumption. These wireless methods have several advantages for the user by reducing the risk of allergy due to the continuous monitoring without direct contact on the user’s skin. In addition, radar systems are not affected by insufficient lighting, unlike video cameras [22,28]. This is an important advantage for the discreet detection of human activities and behaviors, such as falls in environments, like bathrooms, where the risk is higher due to slippery surfaces [29].

Deterioration of the health including progression of an existing disease, loss of cognitive ability and autonomy often result in changes in daily habits and activities. Radar technology used as a monitoring method can monitor walking parameters in a non-invasive manner. In addition to CW radars [30], frequency modulated CW (FMCW) radars [31] can also be used to detect human activity. However, for fall detection and monitoring applications, the complexity of manufacturing, signal processing, and the high cost of FMCW radar do not make it a good candidate for our final application. Moreover, a simple CW radar does not render target range information while on the other hand, an UWB provides distance based on arrival and tends to achieve a better compromise between cost, acquisition performance, and detection capability by using high-resolution ranges.

Recent researches are then focused on advancements and developments of UWB systems for their advantages and feasibility for monitoring gait and fall parameters. The authors in [32] suggested a relation between cognitive functions in the elderly and the gait velocity measured by a micro-Doppler radar. The use of an UWB technology to detect falls in domestic environments is presented in [33]. They compared the performance of supervised and unsupervised fall detection algorithms based on data from the sensors. In [34], some researchers proposed the use of UWB micro-Doppler radar for tracking the walking of more than one person. The study presented in [35] included the use of pulsed Doppler radar to estimate the step time as well as the walking speed. A shortcoming is that most previous works in the literature qualitatively discusses the consistency between radar measurements and recorded motion capture data or vision-based technology, by measuring the gait velocity during a walking test only. In fact, most of their suggested methods are not directly applicable to sequential activities such as sit-to-stand and walking, walking and turning, etc., which better highlight gait and balance impairments. When applicable to the walking activity in straight line, these studies often involve the use of a complex process for measuring the gait velocity parameter. Moreover, the recent research works in the field usually compares gait parameters values without a formal evaluation of a risk of falling level using clinical tests.

In this paper, we therefore introduce a new approach, which leverages recent advanced in these fields. Instead of a simple comparison of performances of sensors with contact and without contact [30], our work focuses on the development of a new combined system for optimizing fall monitoring devices, in order to optimally detect positions, activities, and predict the risk of falls of a patient. We propose to use an UWB radar sensor, a promising technology for discrete home gait analysis, and an existing tool such as an instrumented insole. To our best knowledge, by integrating these two devices, our study makes the first step to investigate the possibility for evaluating without contact a risk of falling in clinical tests and with more than one participant at the same time. This will reduce public health costs and significantly lower the tasks of healthcare professionals. Our proposed approach uses a clinical test such as the Timed Up and Go (TUG) test. This test is comprised of basic everyday movements: stand up from a chair, walk 3 m, turn around (180°), walk back, and sit down again [36]. It was developed to assess changes in functional mobility in people.

Following the description of the radar system and the wearable sensor, we also present in the next section (Section 2) the proposed methodology including the experimental procedure, the TUG data analysis, a gait parameter computation for the two devices using three approaches, and a risk of falling estimation. The first evaluation shows encouraging results, which are presented and discussed in Section 3. Finally, concluding remarks are given in Section 4.

## 2. Materials and Methods

In the next subsections, we present two non-intrusive devices for risk of falling computation and long-term monitoring.

### 2.1. Contactless Sensor

We considered the XeThru X4M200 development kit manufactured by Novelda AS (Kviteseid, Norway) as a sensing device of positioning without contact. This UWB system contains radar SoC, antenna, MCU, header connectors and other hardware for signal processing and connecting interface, etc. Figure 1 shows the typical UWB radar block diagram, which mainly consists of (1) a pulse generator producing a pulse (an amplitude envelope with a very narrow pulse width in the time domain) signal for transmission; (2) a low noise amplifier (LNA); and (3) a correlation circuit. The oscillator drives the pulse generator with a desired waveform such as a Gaussian envelop and determines the pulse repetition frequency of the system. A pulse of electromagnetic waves from the module X4M200 is emitted for a very short time (a few microseconds) and is configurable within two different bands. The lower pulse generator setting enables transmission within the band 6–8.5 GHz, and the higher setting within the band 7.25–10.2 GHz. This module therefore complies with American and European emission limits, respectively, the certification standards of Federal Communications Commission (FCC) and European Conformity (CE). It uses pulse modulation on the transmitted pulses. This modulation spreads the spectrum of the transmitted signal, but also adds a time signature to the pulse.

Compared to other devices, such as the use of Wi-Fi, Bluetooth, and Mobile Phone, the XeThru radar operates under the noise floor of –41 dBm/MHz, and the exposure at its radiation is therefore far lower [37]. This means that during long-term recordings for home daily monitoring, the radar will not cause any health damage. All information about its architecture and functionalities are shown in [38], where the electromagnetic pulses are radiated through a transmitting antenna T. The reflections travel back and are received and sampled through the antenna R (Figure 1). They depend on the size of the reflecting surface and the distance of the target from the radar. Indeed, the radar captures the reflections of every object in its field of view, and the distance to the object corresponds to the position of the reflection in the radar frame. This distance is computed as presented in [39].

For this study, we first improved the pre built-in algorithms from Novelda, which result in an increasing of the initial range from 5 m to 10 m to record data away from the radar with accuracy of ±10 cm. In fact, certain studies [40,41] investigated the possibility to increase the walking distance in a TUG test up to 5–7 m showing advantages among untreated early people with degenerative diseases including Parkinson’s disease (PD). Thus, we believe that our improved radar system could benefit for PD people monitored at an early stage. The frame rate of this improved algorithm is configured into different frequencies and data were transmitted to PC via USB serial port for post-processing.

### 2.2. Wearable Sensor

An insole technology is used as a body-worn device. It has been developed in the Laboratory of Automation and Robotic interaction (LAR.i) of UQAC for preventing falls [42]. This device counts a set of non-invasive sensors such as a three-dimensional (3D)-accelerometer and four force sensitive resistors (FSRs). The 3D-accelerometer is an ADXL345 located inside an electronic board and attached to the foot as shown in [42]. It is a complete 3-axis acceleration measurement system requiring ultralow power and is well suited to measure the static and dynamic acceleration of gravity in order to detect human falls. It measures acceleration with a high resolution (13-bit) up to ±16 g. The FSRs (FSR402, diameter 13 mm) were used for assessing the force distribution under the foot. Two FSRs were placed underneath the heel pad, one medially and the other laterally. The two others were placed under the first and fifth metatarsals approximately.

Various numbers of FSRs [43,44] up to forty-eight [45] are used in computing gait parameters. Although a risk assessment may seem to be better with a high number of sensors or sensors fusion, the challenge remains an effective risk evaluation with an inexpensive device. Thus, reducing the number of sensors unit will help to reduce the memory size, the power consumption, improve the physical integration of sensors and electronics packaging, and therefore reduce the manufacturing cost of an instrumented insole intended to be used at home. In this line of thought, we investigated in [46] the possibility of reducing the number of FSR sensors from four to three over the insole suggested in [42] during a TUG test. For this current study, an accelerometer system, attached to the ankle, along the lower limb, that uses only the y-axis (the direction of walking) and three FSR sensors are investigated.

### 2.3. Experimental Procedure

To evaluate our proposed system, we asked one of the authors of this study (healthy young adult), that we labeled as “participant” in the rest of the manuscript, to perform a TUG test. The instrumented insole was introduced in the shoe of the right foot. We think that a single insole placed inside the right shoe (usually the dominant foot) could measure the complete cycle of the gait. This also makes it possible to reduce the acquisition cost of the insole. The XeThru radar was placed at distance D_pos_ = 0.5 m behind the chair as shown in Figure 2.

The participant was asked to stand-up from a chair, walk along 3 m, turn around (180°), walk back, and sit down again. Before the recordings, the participant comfortably performed trials. Then, the data were recorded by an android application previously designed in [43]. Fourteen TUG tests were performed by the same participant in a laboratory setting. Participant was given as much time as wished to rest between test, and fatigue did not appear to limit its balance control. During each test, the data are sent in real time from the insole to the Android application via Bluetooth and that from the radar to a computer via USB serial port. All TUG tests were performed at the normal and comfortable speed (without being in a rush) of the participant and without any using of stopwatch.

### 2.4. TUG Data Analysis

The computational flow developed for gait and balance parameters extraction is presented in Figure 3 and Figure 4. It involves two algorithms described in the next subsections.

#### 2.4.1. Position-Based Activities Segmentation

We developed an automated position-based TUG signal analysis algorithm. During the TUG test, the algorithm detects six different points using positioning data of the participant. In order to adequately detect these points, the gait speed is firstly estimated before the TUG activities segmentation.

1.Gait velocity estimation

An accurate estimation of the velocity plays an important role in the identification of the beginning of the sit-to-stand activity or that of the walking. The gait velocity can be obtained by first order differentiation computations of the position measurements. However, the position signals have often some estimation errors. Therefore, the direct differentiation of the position signals tends to increase the noises, particularly in low-velocity and low-acceleration regions. To overcome this situation, several methods summarized in [47] including the use of Kalman filter (KF), a widely used filter in target tracking, have been proposed to reduce the errors and thus improve the accuracy of velocity and acceleration computations. In our study, available information as input for KF corresponds to the position data of the participant during the test. The algorithm of KF is described in [48]. The state of the KF involved the position, velocity, and acceleration. The first equations describe the way in which a future state of the participant can be predicted. It uses values of previous position, previous velocity, current acceleration, and previous accelerometer bias. The second equation is the computation of KF gain. The final equations mainly contain the process noise matrix we denoted by Q_k_ and the measurement noise covariance R_k_ for state’s update at step k. Ideal values of Q and R are difficult to obtain. Nevertheless, approximate behavior, such as the standard deviation of the position data obtained from multiple testing, and compared with true positions, can be used.

As a result, we used $R_{k}=R=0.04$ and for the initial process noise matrix, we initially used $\bar{Q_{0}}=\left[\begin{array}{ccc} 0.04 & 0 & 0\\ 0 & 0.01 & 0\\ 0 & 0 & 0.01 \end{array}\right]$. Since our final application will be to collect data at home for days, the length of the tracking update (the time interval) is set to T_s_ = 0.2 s. However, different frequencies can be used.

2.Segmenting the TUG signal from a radar

As the participant starts the test, the algorithm (Figure 4) accumulates the position data for a duration of 3 s. This duration was chosen according to the normative maximum duration of the sit-to-stand activity in the elderly population [49] and also to deal with possible high values of the position due to the participant movements (hand movements, etc.) in the sitting position. The algorithm then estimates the velocity (V) using the KF method described above. If $\left|V\right|<{\;V}_{\mathrm{thres}\;}\left(V_{\mathrm{thres}}=0.4\;m/s\right)$ in all the 3 s-window, the algorithm continues to accumulate the position data for a certain time X_secs_ = 30 s. The threshold for the velocity is set based on the literature [50] and also experimentally from the gait event. This suggested threshold could be a good candidate to take into account the irregular and slow gait observed in people with PD. However, the end-user can test different threshold values and find the best one based on the walking event detection results. After the X_secs_ duration and if the instantaneous velocity is still less than the threshold, the algorithm considers the data as from a quasi-static movement (sitting with little movements or without movement, or remain in standing posture, etc.) and stops the process. However, if $\left|V\right|\geq V_{\mathrm{thres}}$ the algorithm searches in the last 3 s of position data (D): the first local point where D > D_pos_, which corresponds to the point T_0_, and the first local point where D ≥ (D_t1_ = d_qs_ + D_pos_) starting from T_0_, which corresponds to the point T_1_. These two points correspond, respectively, to the beginning and the end of the sit-to-stand activity from the TUG test (see Figure 2).

The stages of the algorithm are summarized in a flow diagram (Figure 4). Once the position *D* becomes equal to D_t2_ = 3 m + D_pos_, the system partitions the first region, which corresponds not only to the walk forward (WF), but also to the beginning of the turning phase (point T_2_). The algorithm continues searching in the available data position. When it finds that *D* is equal again to the 3 m + D_pos_, it starts to partition the walk back (WB) region where its beginning corresponds to the point T_3_. The turn activity is deduced as the data between T_2_ and T_3_ (see Figure 5). Figure 5 shows an example of the radar position signal of a healthy control subject performing a TUG test. This example highlights a case in which the participant sat in the chair for a certain time before standing up and starting to walk. For a personalized analysis and for home usage, the threshold of the velocity (V_thres_) and the distance at which the radar is placed (D_pos_) can be adjusted by physicians, clinicians, or domain experts to tailor the daily activities segmentation included in the TUG test.

After this segmentation, the algorithm can finally compute the following gait and balance parameters (presented in Table 1): (a) stride length (SL) defined as the distance between successive points of initial contact point of the same foot; (b) stride time (ST) defined as the time from initial contact of one foot to initial contact of the same foot; (c) cadence is the number of steps taken per second. Thus, the instantaneous cadence can be calculated as 60 times the number of steps taken in one stride divided by the stride time. The number of gait strides is calculated as the number of the right heel-strike points; (d) stride speed (SS) or stride velocity defined as the product of SL and the inverse of ST.

#### 2.4.2. Acceleration-Based Activities Segmentation

Using the y-axis of the acceleration (Ay) from the insole, we developed a second algorithm to automatically segment the TUG test into different activities. The algorithm first normalizes the signal acceleration by subtracting the mean and dividing by the standard deviation. The total force (sum of the three FSRs) signal was normalized by its maximum. To detect the TUG phases, the Ay signal is low-pass filtered successively by a median and a Butterworth filters.

The form of the acceleration Ay and the total force signals (Figure 6) allow detecting the number of steps performed by the participant. During the walking (Figure 6a), it is known that a temporal signature coming from measurement of acceleration signals is repeated at each step [51]. As a result, it has been observed by looking at this temporal signature that the signal crosses the zero level twice in each step. We note that when the foot is totally resting on the soil, the acceleration components are constant and can be around zero. In our study, the number of strides was taken as the number of correct acceleration peaks signal minus one (the correct peaks are shown in Figure 6b). To detect the number of peaks in Ay (Figure 6a), a threshold with a range between 0 and 1 can be set [42,52]. This threshold can be defined as the ratio of the maximum value of the acceleration. For example, 0.5 indicates that the threshold is set to 50% of the maximum of the anteroposterior acceleration (acceleration along the y-axis) [50]. When a more irregular form is analyzed, for example in the case of persons with PD, the threshold should be less than 0.4 [50]. However, contrary to most studies in literature, our step identification is based on the shape similarity [53] rather than the use of thresholds.

To detect steps and also to avoid missteps detection, the so-called blocking period (lockp) and zero crossing detector described in [53] have been developed. This method calculates the average time between the zero-crossing detected so that a locking period can be set as a moving window (Figure 6a). In each window, the maximum (peak) acceleration is detected. Afterwards, a new window (around the peak) equal to the size of the locking period is set. Within this window, differences between a peak and the minima before (defined as $P_{L}$) and minima after (defined as a $P_{R}$) were calculated (Equation (1)). When the difference between the current peak and the next peak (labeled diff_max) does not respect a predefined range time (Equation (2)), the current peak is set to zero (Figure 6b).
(1)$$\left\{ \begin{array}{l} P_{L}=\max_{i}-\mathrm{val}(\mathrm{TO})\\ P_{R}=\max_{i}-\mathrm{val}(\mathrm{HS}) \end{array}\right.$$
(2)$${\mathrm{lock}}_{P}\leq \mathrm{diff}\_\max\leq \mathrm{Max}\_\mathrm{time},$$
where $\max_{i}$ represents the maximum (peak) value of the acceleration detected in the window i; TO: toe-off; HS: heel-strike (Figure 6a). $P_{L}$ is the difference between the value of the peak (max_i_) and the value of the minima before, i.e., the minima at left (val(TO)) whereas $P_{R}$ is the difference between max_i_ and the value of the minima after, i.e., the minima at right (val(HS)).

Step counts in acceleration signals were validated against the steps counted in the force sensor signals in which the maximum and minimum of the force (the stance and swing phases of gait) are considered, giving a more robust algorithm. In fact, during a walking phase, the maximum pressure value from FSR sensors is recorded during the stance phase, i.e., when the foot reaches horizontal and the fifth metatarsal touches the soil.

It is known that the speed to turn is linked to the progression of PD [54]. Thus, this phase is relevant to distinguish more efficiently people at the beginning of the PD. Despite the importance of this phase, it was not evident to segment and isolate this region with an accurate precision using the acceleration or the FSR sensor. Nevertheless, using Ay, we hypothesize that the maximum pitch performed during this phase would be a complete walking cycle (one stride), except in the case of freezing of gait or festination where the person with PD can perform several small consecutive steps or no step. Figure 6 mainly shows the segmentation of the TUG test into different strides for the walking phase (as highlighted in Figure 6a,b).

The beginning of the sit-to-stand causes a rapid change of the body center of pressure (CdP) displacement in the anterior-posterior direction which is reflected on the FSR sensor data (Figure 6c). This phase starts generally with the heel-strike where the pressure in the center of the heel starts to increase. At the end of the sit-to-stand activity, the force sensors under the feet reach their highest value in the standing posture. The stand-to-sit activity is marked by a decrease in the force measured under the foot from a maximum value. After this segmentation, in different activities of the TUG test, the algorithm can finally compute the gait and balance parameters, as presented in Table 1.

### 2.5. Comparing Radar and Instrumented Insole

Optimizing the number of gait and balance parameters used in the evaluation of a risk of falling is still a challenge. However, as summarized by Roberts et al. [55], the gait velocity, cadence, stride length and step length were those parameters most frequently computed in the literature. In addition, previous works have shown that these parameters have good to excellent test-retest reliability [40,54]. Furthermore, previously studies found that these parameters, especially the stride velocity, are the most sensitive variables particularly in early persons with PD [41]. Thus, in our study, these parameters are computed (Table 1) for testing the proposed system. Indeed, stride velocity prove more capable of discriminating between old versus young adults [56] or gait quality between older fallers and non-fallers [57]. Therefore, we chose to compare the two proposed devices by using the stride length (SL) which is linked to the stride velocity.

#### 2.5.1. Stride Length Computation

The classic existing stride length estimation method, such as the use of double integration of the acceleration, can work mostly in cases of walking in a straight line [44]; however, some computation errors can occur rapidly due to the noise in the sensor or due to the type of activities, such as a turning activity. Thus, having an accurate stride length estimation is a fundamental component. Other methods, such as biomechanical methods, statistical regression, neural network, etc. [58,59], can also be used. However, most of these methods available in the literature imply a rather cumbersome process to implement and/or require learning data process. Therefore, since our proposed system should reach a wide commercial range available in the retail trade, we suggest using a statistical computation method involving calibration procedures to assess a stride length parameter. In the following, to get a better estimation of this parameter for the insole, three different approaches are used by comparing them with one provided by the radar. By doing so, all proposed algorithms in this study, as they are based on simple statistical computation statements, without any recursion do not demand complex computational time.

1.First approach

The first approach in Equation (3) is proposed by Kim et al. [60]. This equation represents the relation between measured average of acceleration and stride length (SL):(3)$$\mathrm{SL}=K_{1}*\sqrt[{3}]{\sum_{i=1}^{N}\frac{\left|A_{y}^{i}\right|}{N}},$$
where $A_{y}^{i}$ represents the value of the acceleration along the y-axis (direction of walking) at each point i, $K_{1}$ is a constant for unit conversion and N is the total number of samples.

2.Second approach

The second approach is based on the Weinberg algorithm [61]. The algorithm assumes that SL is proportional to the vertical movement of the human hip. This hip bounce is estimated from the largest acceleration differences at each step. SL is calculated using the empirical nonlinear model as shown in Equation (4):(4)$$\mathrm{SL}=K_{2}*\sqrt[{4}]{A_{y}^{\max}-A_{y}^{\min}},$$
where $A_{y}^{\max}$ and $A_{y}^{\min}$ represent respectively the maximum and minimum of the y-axis acceleration and $K_{2}$ is a constant for unit conversion.

3.Third approach

Scarlett [62] proposes another approach, a simple algorithm to determine the SL, as presented in Equation (5). This third approach showed that there is a correlation between the maximum and minimum values, average of acceleration and step length:(5)$$\mathrm{SL}=K_{3}*\frac{1}{N}\sum_{i=1}^{N}\frac{A_{y}^{i}-A_{y}^{\min}}{A_{y}^{\max}-A_{y}^{\min}},$$
where $A_{y}^{i}$ represents the value of the Ay (acceleration along y-axis) at each point i, $A_{y}^{\max}$ and $A_{y}^{\min}$ represent respectively the maximum and minimum of the Ay, N is the total number of samples, and $K_{3}$ is a constant for unit conversion.

These constants for unit conversion are generally determined experimentally as they need to be calibrated for each user and each walking pace [58,60,63,64,65]. Based on the literature and the different experimentations, in this study, we used $K_{1}$ equal to 0.98 [60]; $K_{2}=K$ and $K_{3}=2*K$ where K is a calibration coefficient obtained by the ratio of estimated distances and real distances. In order to get the value of K, we examined 56 sets of data obtained at different stride length labeled from 1 to 4 (Figure 7). The Figure 7 displays the mean and the error bar. Each symmetrical error bar represents the standard deviation of fourteen tests.

#### 2.5.2. Risk of Falling Analysis

The main concept of the risk of falling adopted in this manuscript is based on the different methods (including the use of gait deviation index) proposed in the literature [44,46,66,67,68,69,70,71]. Most of them cannot be easily interpreted by nonprofessionals since they have high index above 100 without specific interpretation. In this section, we investigated a new risk of falling index for home settings (Figure 3) based on the computed parameters (Table 1), notably their variabilities. These variabilities depend on the height, weight, age, sex, and several other physiognomic factors. Since these parameters vary considerably from one person to another, the computation of this variability is often carried out relative to a reference. This reference or baseline can be the average of several steps of the participant on a regular soil under ideal conditions, when the person is walking normally, or can be the average of three or more tests (as example: three TUG tests). In the case of a people with a gait disorder or neurological disease, it would be more appropriate to use a reference from healthy people. The proposed risk index is summarized in Equation (8), where its computation process follows three stages:

(1) From the value of the current gait parameter (labeled $G_{k}$), the algorithm firstly computes, for each selected parameter k, a ratio $R_{k}$. This ratio as shown in (6) is how many standard deviation the value is from its average $G_{k}^{M}$:(6)$$R_{k}=\left|\left(G_{k}-G_{k}^{M}\right)/\sigma_{k}\right|,$$
where k corresponds to the gait parameter and $G_{k}$ its value; $G_{k}^{M}$ and $\sigma_{k}$ are respectively the mean and the standard deviation of the values $G_{k}$ computed from the TUG tests (the baseline). Among the same participants, the average is computed, taking into account multiple strides from multiple tests performed before testing. This average can also come from control subjects.

(2) In the second stage, we adopted the idea suggested by Schwartz [66], which find many applications in the literature [67,68,72,73]. It consists of multiplying a ratio by 10 and subtract them from 100 to give at each stride a gait deviation index we labeled $S_{k}$ (Equation (7)):(7)$${\;S}_{k}=100-10\times R_{k},$$
where $S_{k}$ is the suggested risk of falling (ROFA) score according to the gait parameter k. Since we are investigating a score that can be easily interpreted by a user, unlike the study in [66], we propose an index that does not exceed 100. Therefore, each value of this index could correspond to a level risk, for example a risk level setup at 1, 2, or 3.

This score (Equation (7)) is then computed between 0 and 100 and can interpret as follows, where more interpretations can be found in [67]:

${\;S}_{k}$= 100 indicates the total absence of gait pathology. It means that the gait of the participant is close to the average gait parameters from the control subjects.Every 10 points that the $S_{k}$ falls below 100 corresponds one standard deviation away from $G_{k}^{M}$ or away from the average computed in the control subjects.

(3) Thereafter, in the third stage, each $S_{k}$ multiplied by equal weight $W_{k}$ (depend on the number of gait parameters ${\;N}_{k}$), is then summed together. The result computed in Equation (8) gives an indicator for the risk of falling (ROFA) during the TUG testing:(8)$${\mathrm{TUG}}_{\mathrm{score}}=\;\overset{N_{k}}{\underset{i=1}{\sum }}W_{k}{*\;S}_{k}$$

In fact, if the current stride data is close to the computed average data, the ratio $R_{k}$ will tend to zero and the actual value (the ROFA score) will be high indicating a low risk. Based on the literature [70,74], this TUG score can be interpreted in three, four, or five levels. For this study, we suggested the interpretation as follows:0 to 24 indicates a very high fall risk;25 to 49 indicates a high fall risk;50 to 74 indicates a medium fall risk;75 to 99 indicates a low fall risk;100 indicates a very low fall risk.

#### 2.5.3. Statistical Analysis

To evaluate the accuracy of the insole to estimate a stride length, we used the Wilcoxon signed rank test, a non-parametric, paired, two-sided test for the null hypothesis ($H_{0}$) since we are comparing two groups in the sample:(9)$$\left\{ \begin{array}{l} D_{i}=X_{\mathrm{Ins},i}-X_{\mathrm{Rad},i}\\ H_{o}:M_{D}=0 \end{array}\right.,$$
where $M_{D}$ is the median of the population of difference $D_{i}$; $X_{\mathrm{Ins},\;i}$ and $X_{\mathrm{Rad},\;i}$ are the set of the stride length values, respectively, computed by the Insole (Ins) and the radar (Rad); i represents the number of strides acquired during all the TUG tests performed in the experimental protocol, as described in Section 2. A statistically significant is set at the 95% confidence level (*p* < 0.05).

## 3. Results and Discussion

The main goal of this pilot study is to relieve congestion in medical settings by investigating the feasibility of administering a TUG test thanks to an UWB radar sensor. For doing so, we investigated the main contribution in different aspects.

### 3.1. Sensors Reduction Process

The system we propose should reach a wide commercial range available in the retail trade. For doing so, in Figure 8, we are optimizing its function by investigating the reduction of the number of sensors used and therefore the optimal number of gait and balance parameters necessary for ROFA evaluation. The present results suggest the potential utility of employing only two FSRs sensors during walking testing to identify gait impairments by the evaluation of the ROFA (Figure 8). The two FSRs should be placed at specific location, i.e., one FSR at heel and the second at toe locations. This may allow reducing the power consumption, memory size, manufacturing cost, and improve the physical integration of sensors and electronics packaging. As shown in Figure 8, it is important to emphasize that two FSRs provide statistically similar information (*p* > 0.05) with the y-axis acceleration for computing the same stride time, and, therefore, the corresponding stride data segmented. This means that if the radar is supplemented with the insole, a reduced number of sensors could be used on the wearable sensor since the radar can be used to compute a stride length. However, the final number of force sensors mainly depend on the end-user application.

### 3.2. TUG’s Activities Segmentation for Enhancing Gait and Balance Disorders Detection

Figure 9 shows the output of the proposed system showing both the segmentation of the TUG test in different activities using respectively the radar and the instrumented insole.

Since these two devices have different frequencies sampling, we used a classical interpolation method, such as the cubic data interpolation [75] on the data from the radar (Figure 9). However, the transition points (T_i_) are firstly determined in the original data (as presented in Figure 5).

Figure 10 compares the performance of a radar and an instrumented insole to estimate a stride length during walk and turn activities. In Figure 11, we also emphasize that the use of two FSRs may provide statistically similar information for computing the ROFA levels compared to the use of three FSRs during a walking activity. While the first system (use of an instrumented insole only) can assess a risk of falling with only one participant at time, the second system (use of insole and radar) could present more benefit since an UWB can track multiple participants at the same, and probably, in doing so, evaluate falls in more participants at the same time.

### 3.3. Contactless TUG Testing

Since the instrument cannot be worn at all times, an automatic risk of falling evaluation can become more difficult due to the fact that a risk, owing to a spasmophilia, physical asthenia, freezing of gait, tremor, etc., may occur at any time among people with more severe motor impairments. In this study, we therefore further explored the possibility of using a non-contact device for a clinical test such as the TUG test. Our results firstly show contactless TUG testing is feasible and that the position data from a radar can be enough to identify the different activities included in such clinical tests (Figure 5 and Figure 9).

It is known that our neural systems control different aspects of gait and mobility [76]; however, the brain mechanisms causing a variety of gait impairments have not yet been fully translated into the clinical gait parameters, which may explain the increasing number of falls, as example, during a sequence of activities (walking and turning, stand up from the toilet and walking a few steps, etc.) [29]. This increasing number of falls could be due to the portable equipment available for quantifying the different daily activities, which is supposed to highlight the human performances any time as much as possible. Indeed, compared to the radar, wearable sensors, such as marker-based provide equally results of the walking parameters [30]. However, their proposed method may not only confuse the turning task, but may also be unsuitable for long-term wearing and discretion. Contrary to the literature, this current study therefore proposes to employ two unobtrusive and non-invasive devices (radar and instrumented insole) where one can easily supplement the other or replace it in order to provide continuously more accuracy measures of the human performances. By using a contactless sensor, the more global approach we are developing here is to fully link and efficiently the human activities such as walking, turning, etc. [52,77], to the human factors, such as the Hoehn and Yahr scale, state of neurological disease, taking medication, adjusting drugs prescription, etc. [43], for a long-term monitoring.

### 3.4. Effect of the Turning Task

It is well known that the 180 degrees (180°) turn performed at the 3m of the TUG test (Figure 2) is a best candidate for evaluating balance capacity [78]. This turn is often used in rehabilitation and contains a gait pattern change to highlight any mobility difficulties. Therefore, secondly, by exploiting the radar technology, our findings showed a potential for segmenting more easily and efficiently the 180° turns as part of a balance and stability (Figure 9). The number of steps during this phase was count using the insole technology (Figure 6 and Figure 9). The results indicate that turning activity could have an important role in the balance performance and show that neural systems impact not only the gait parameters such as the stride length, but also depend on the type of activity (Figure 10 and Table 2). Since the gait variability worsens as disability increases, it has been important to investigate in this work the variability of gait in different contexts (Figure 10). Indeed, some studies showed that change in certain gait parameters may be predictive of future falls and physical functional decline [79]. By exploiting the proposed unobtrusive and non-invasive system, some gait parameters, such as cadence, stride length, stride time, stride velocity have been computed. The next subsection only discusses the stride length outcomes as this parameter is used for comparison.

### 3.5. Discussion on the Stride Length Computation

In biomechanical research, the measurement and analysis of gait parameters reveal information about an individual’s health and aid in the diagnosis and treatment of patients. Contrary to previous studies [32,34,35,80,81], we focused on the stride length measurement during a Timed Up and Go (TUG) test using three different approaches. Step length and stride length (twice the step length) are two important measurements in many applications such as gait analysis and human activity recognition. As shown in [82], the increased fear of falling score was associated with increased stride length variability, and compared to heathy adults, people with PD show a reduction in their stride length [83].

Thus, thirdly, using the acceleration data, three different approaches are tested to make it ensure that the insole can be well exploited for stride length estimation. As results, there was no difference when compared walking forward (WF) and walking back (WB) using the first approach (*p* = 0.0803 > 0.05), second approach (*p* = 0.1677 > 0.05), and third approach (*p* = 0.6848 > 0.05). These findings show how the three approaches can perform well when the human is walking in straight line without turning. However, a significant difference (*p* < 0.05) was found by evaluating the performances of these approaches against the radar’s estimation during the WF or WB (Table 2). As stated previously, a sequence of activities is most important for evaluating a risk of falling in elderly. We therefore analyzed the part constituted of the combination of one stride before turning, turning phase and one stride after turning (denoted as B and A Turn). Considering this part, and compared to the radar, the first two approaches were most sensitive (*p* < 0.001) contrary to the third approach (*p* = 0.2477 > 0.05). The stride length extracted from the straight-line (WF or WB) and B and A turn are then different, but were in excellent agreement compared to the previous studies reported in [79]. These differences may result in the fact that, contrary to the walking in straight-line, neural systems related to turning task may be more vulnerable showing more gait impairments. Moreover, these different outcomes can be due to the variety of motion patterns during walking as well the location of the sensor on the foot. However, in the global process (combining data from walking and turning denoted as Walk (W) and Turn in Figure 10), the third approach provided better results with a root mean square error (RMSE) equal to 0.3675. Furthermore, it was non-significant compared to the radar estimation (Table 2). Since the best location of the sensor on the human body can be hard to solve, we think that the use of radar can replace the wearable technology for better estimation of the stride length or stride velocity. Indeed, our main findings showed that when a radar is used in the TUG test, participant walked with a similar but accurate stride length compared to the one computed by the insole technology. It was significantly shorter than the one computed by the insole except for the third approach (Figure 10, Table 2). The Figure 7 and Figure 10 showed that the stride length estimated by the radar is relatively close to a true position, and when its value is used in ROFA assessment, this could enable the possibility of reducing the number of sensors available on the insole and reflect more the level of the balance problem.

### 3.6. Clinical Implications

By enhancing the existing methods, our goal is to increase the quality of life of the users, and significantly lower the tasks of the caregivers when tested multiple participants. The suggested TUG test can be performed at home or residence for elderlies and can be analyzed automatically and remotely using our suggested methods. Component-based TUG can be processed combining the radar with either the acceleration or two force sensors. The radar combined with an insole may provide additional information on the risk evaluation during the TUG test (Figure 11), e.g., when the subject turns, or when the energy of the insole is low, etc. The use of the radar makes the proposed method much easier to be applied in practice especially in bathroom or in other settings where the falls are important. Our proposal can possibly help doctors to monitoring the rehabilitation of multiple elderlies remotely and automatically on a regular basis. Since we are interested in the long-term monitoring, by involving the radar, the proposed system can be used to assess the vital signs, the progression of a disease and the improvement between the clinical visits and thus help to decrease the number of visits to physicians and clinicians. In addition, it can give information to the neurologist to adjust drug prescription as needed.

### 3.7. Limitations of This Study

The limitation of this study is the generalization of finding to a wider population, due to the small sample size used. However, this was a pilot study as a proof of concepts. It is not likely that the health status of the participant is the cause of our results because we are focusing on a non-contact system for assessing daily performance at home. However, more works should confirm our findings in larger groups. Thresholds for gait velocity and constant K are currently defined and based on experiences. In the future, we want to investigate these thresholds according to more participants’ individual capabilities in order to make the computations more reliable in larger sets of participants. The algorithms based on the radar data for activities segmentation could be further enhanced for people with more gait disorders. Our future works will address these issues as well as investigate more the 180° turns, the sit-to-stand and stand-to-sit activities, and the vital signs from the UWB radar in a more realistic daily living setting with a larger population including elderlies. However, the combination of data from two FSR sensors and one single radar is rather simple at this moment, and show encouraging results for a clinical test, such as a TUG test.

## 4. Conclusions and Future Works

In this paper, we proposed an UWB radar sensor for the risk of falling analysis and exploited three different approaches for computing a stride length during a clinical test. We evaluated the accuracy of these approaches on the instrumented insole data and compared their performance against the radar outputs. A statistical comparison showed a non-significant effect, especially when the walking is in straight line without turning. Moreover, for a full dataset from a TUG test, results showed that the third approach has a lower RMSE than the others, indicating the usefulness to investigate the method when computing a stride length value for a wearable sensor, such as an instrumented insole. In future work, rather than performing a TUG test, a continuously monitoring of vital signs, gait, and turning in normal daily activities could be performed at home using the radar technology in order to acquire large amounts of data throughout the day. More research should be done to determine the environmental factors and human characteristics that influence the gait and mobility in both healthy people and people with neurological disorders.

## Figures and Tables

**Figure 1 sensors-21-00722-f001:**
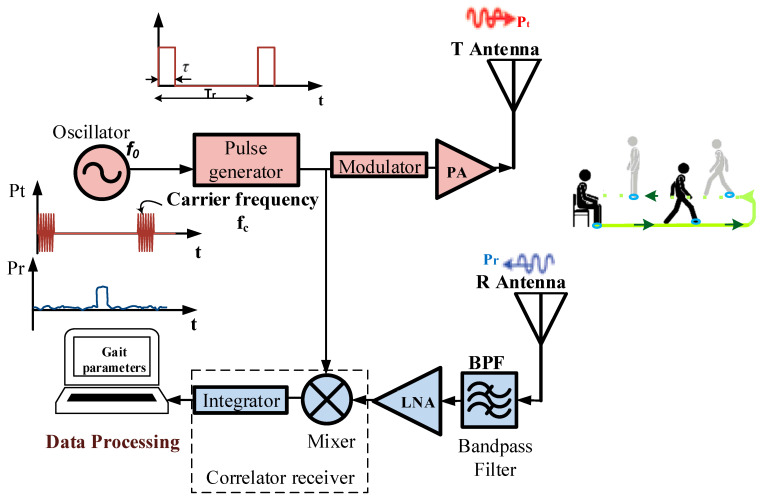
Ultra-wideband (UWB) radar block diagram.

**Figure 2 sensors-21-00722-f002:**
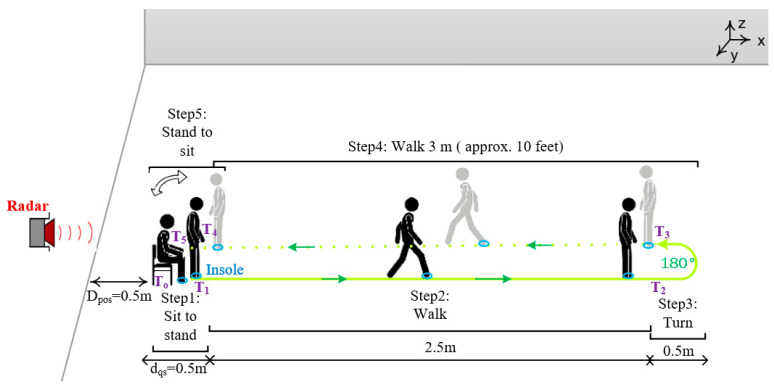
Experimental setup for home daily risk of falling evaluation using UWB radar and instrumented insole. (D_pos_: the distance between the location of the radar and that of the chair; d_qs_: the distance between the chair and the person in standing position, which correspond to the sit-to-stand activity).

**Figure 3 sensors-21-00722-f003:**
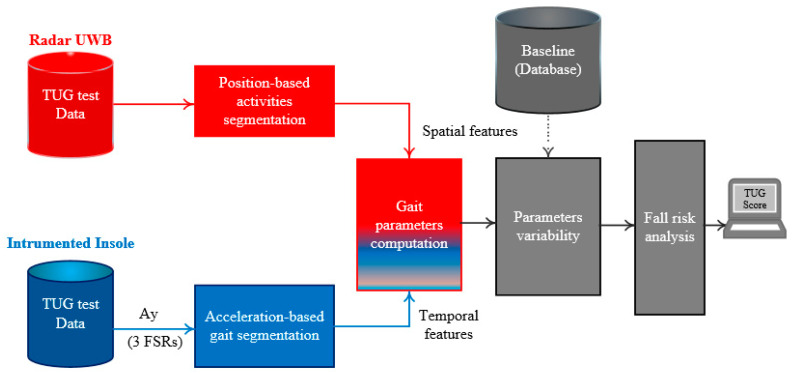
Computational flow of the proposed method for evaluating a risk of falling in home settings using a TUG test. (Ay: the y-axis of the three-dimensional (3D)-acceleration; TUG: Timed Up and Go).

**Figure 4 sensors-21-00722-f004:**
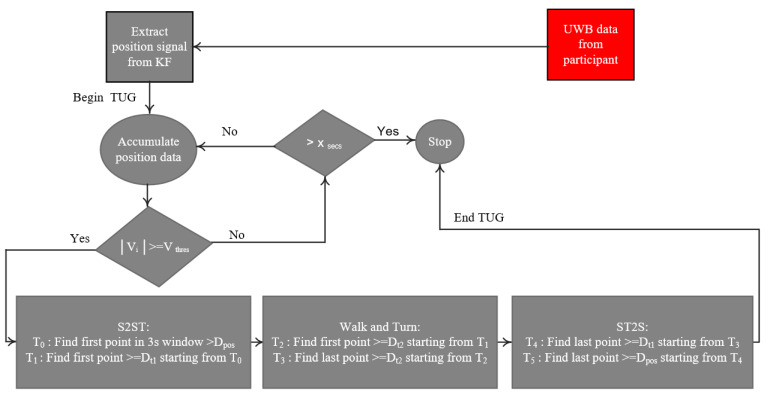
The flow diagram of the automated algorithm for segmenting TUG activities using the UWB radar signal. (D_pos_: the distance between the location of the radar and that of the chair; KF: Kalman filter; V_i_: is the instantaneous gait velocity; S2ST: Sit-to-stand; ST2S: Stand-to-sit; T_i_: different points detected for TUG segmentation; D_ti_: predefined distances during the TUG test).

**Figure 5 sensors-21-00722-f005:**
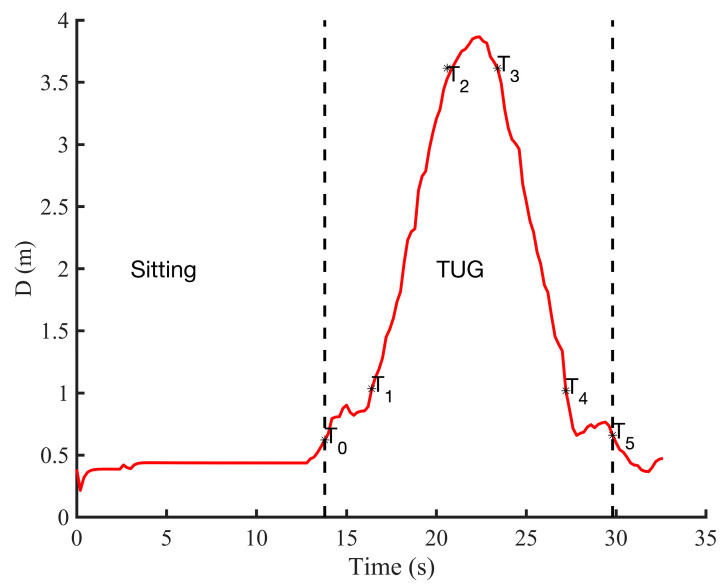
An example of the positioning data (distance D in meters) from a testing performed by a healthy young adult showing the different points detected for TUG activities segmentation. TUG: Timed Up and Go.

**Figure 6 sensors-21-00722-f006:**
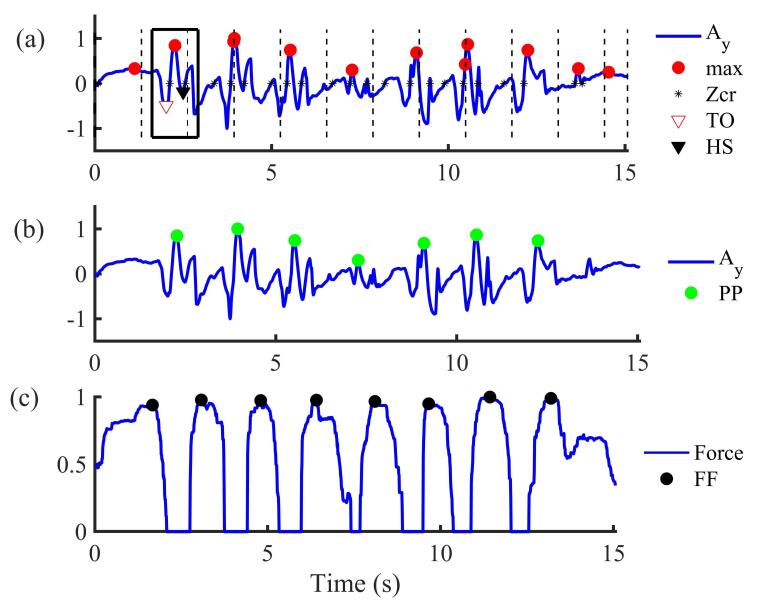
Insole-based gait segmentation in different strides: (**a**) Initial acceleration data from the insole without applying the suggested algorithm (Ay: acceleration in y-axis; Zcr: zero-crossing; TO: toe-off; HS: heel strike); (**b**) Acceleration data after applying the suggested algorithm (PP: peak-to-peak for strides detection). The Stride is therefore the difference between one green peak and the second next green peak; (**c**) The total force from the three FSRs (FF: foot flat). The square in solid line represents the moving window in which its length is equal to the locking period; the red circle is the original peak detected and the green circle indicates the correct peak detected after applying our proposed algorithm.

**Figure 7 sensors-21-00722-f007:**
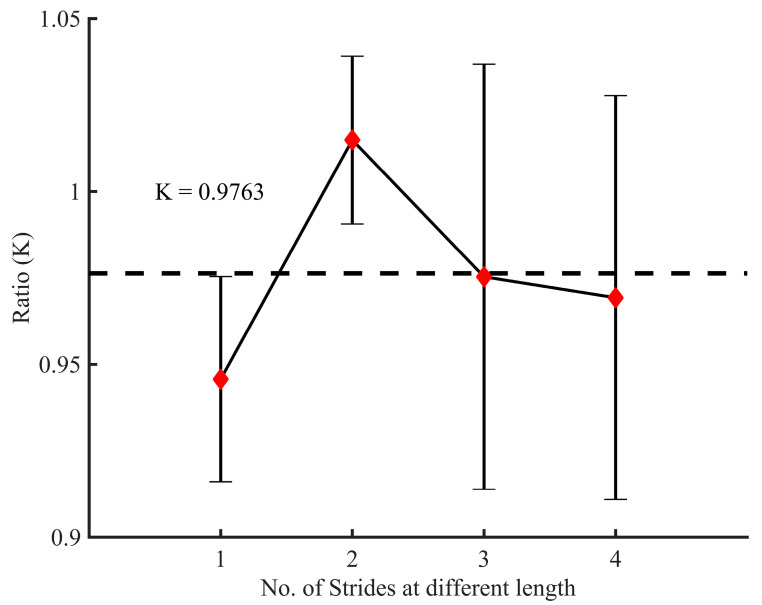
Fourteen experimental tests for calibrating the constant K in the different approaches. The ratio K is equal to the estimated distances from the radar divided by the real distances measured. The dotted line represents a mean where K = 0.9763.

**Figure 8 sensors-21-00722-f008:**
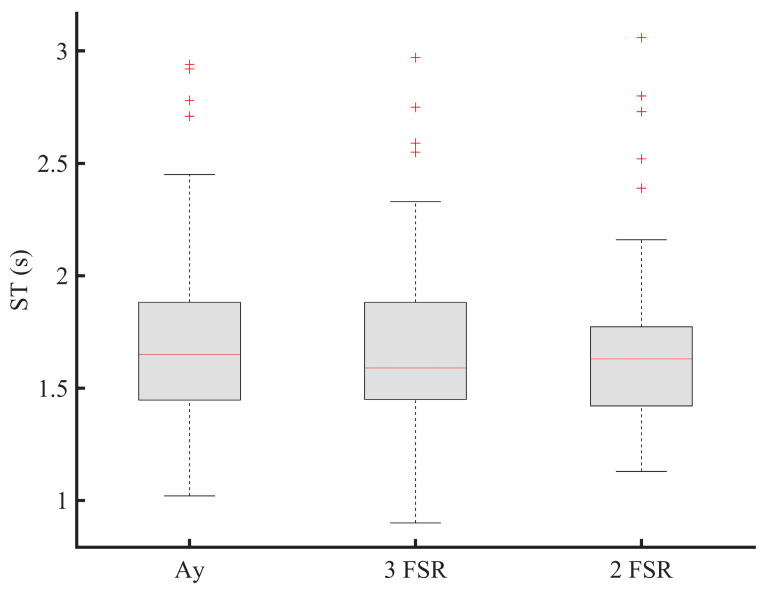
Stride time (ST) estimation for sensor reduction (Ay: y-acceleration; FSR: force sensitive resistor; 3-FSR: use of three FSRs; 2-FSR: use of two FSRs).

**Figure 9 sensors-21-00722-f009:**
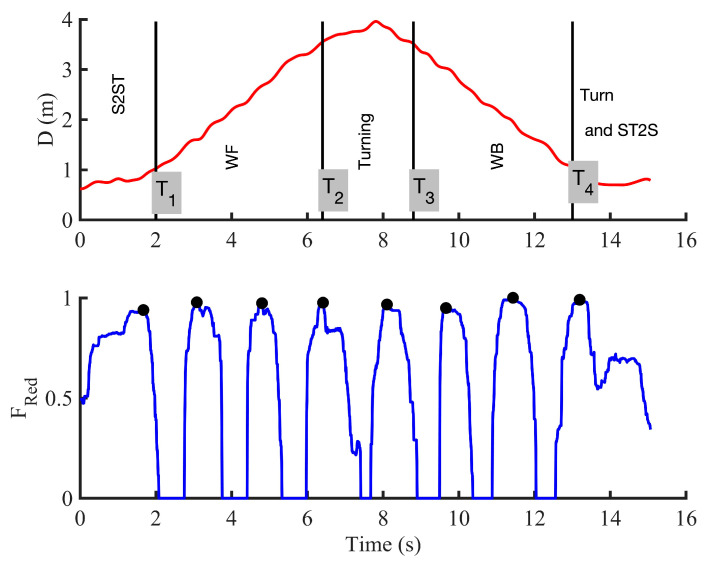
Proposed system showing the segmentation of the TUG test in different activities using the radar and the instrumented insole. The walking phase in this test was segmented in different strides using only two FSRs. (S2ST: sit-to-stand; WF: walk forward; WB: walk back; ST2S: stand-to-sit; D: distance measured by the radar antenna; F_red_: use of two FSRs from the reduction process).

**Figure 10 sensors-21-00722-f010:**
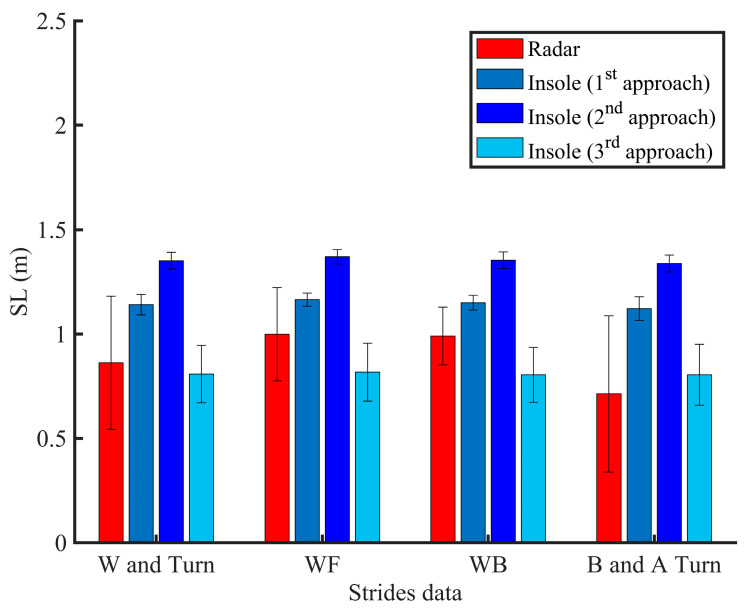
Stride length (SL) estimation during the TUG test using different approaches and technologies (W: walking; WF: walk forward; WB: walk back; B and A Turn: strides before and after turning).

**Figure 11 sensors-21-00722-f011:**
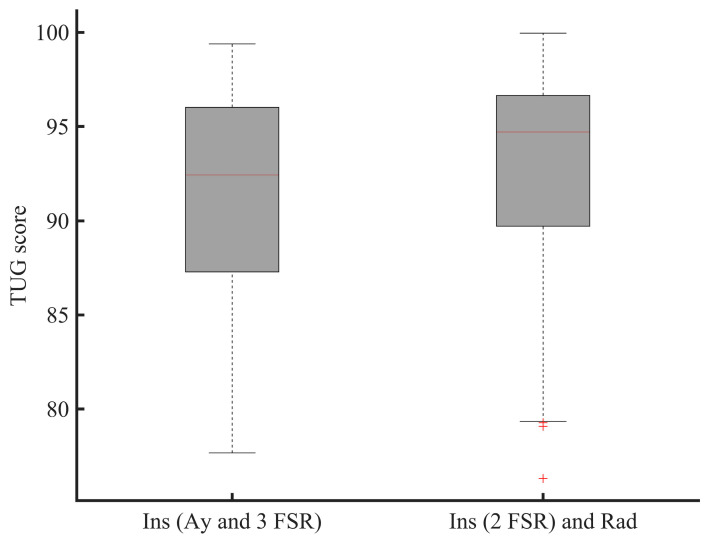
The TUG score of the participant performing the TUG tests. The data from the walking and turning phases of this test are exploited (FSR: force sensitive resistor; 3 FSR: use of three FSRs; 2 FSR: use of two FSRs).

**Table 1 sensors-21-00722-t001:** Gait parameters computed by the instrumented insole and the radar during the TUG test.

System	Parameters Measured by the System
Insole	Cadence—Stride time—Stride speed—Stride length
Radar	Stride length—Stride speed

**Table 2 sensors-21-00722-t002:** *p*-values of the tested approaches using a pairwise Wilcoxon signed rank test and the root mean square error.

Methods	Radar Estimation vs. Approach Estimation			WF vs. WB (*p*-Value)	Walk and Turn	
	WF (*p*-Value)	WB (*p*-Value)	B and A Turn (*p*-Value)		*p*-Value	RMSE
1st approach	0.0215	0.0061	<0.001	0.0803	<0.001	0.4099
2nd approach	<0.001	<0.001	<0.001	0.1677	<0.001	0.5761
3rd approach	0.0171	0.0024	0.2477	0.6848	0.2083	0.3675

Notes: WF: walk forward; WB: walk back; B and A turn: before and after turn; RMSE: root mean square error.

## Data Availability

The data presented in this study are available on request from the corresponding author. The data are not publicly available due to privacy and ethical restrictions.

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
