# Peer review of "Risk of Falling in a Timed Up and Go Test Using an UWB Radar and an Instrumented Insole"

_sensors, 2021, doi:10.3390/s21030722_

Round 1
Reviewer 1 Report
The paper presents a goal-oriented study that aims at creating a complex monitoring system.
Drawbacks and remarks:
- Equations 3, 4 and 5 look senseless in the form they are presented in paper. It is evident that, in each of them, the dimension of the variable written in the left-hand died DOES NOT correspond to the dimension of the expression given to the right of equation sign. SL is basically length, which is expected to be measured in meters, whereas the dimension of the expressions is far from it, i.e., cubic (eq. 3) or fourth (eq. 4) order roots of acceleration (m/s2) or even unitless (eq. 5). To make things worse, the same dimensionless coefficient K presents in all of the equations.
- The main concept of falling risk adopted in the paper seems rather misguiding.
- The formula itself contains terms which are ambiguously described afterwards: are those three terms Sk, Sk, sk different or mean the same?
- It is not clear what is actual value (related to test run, frame, stride, etc.?) of the parameter and what is its mean value.
- The most important, the parameter evaluated in eq. 6 seems to be some sort of deviation observed for some parameter with respect to its mean value. However, there is no explanation given in the present paper how this deviation relates to the falling risk. Moreover, nothing about the falling risk is said in the ref. no. 56, where the similar index has been introduced. As long as risk falling is even the part of the paper title, it must be looking weird that any metrics related to it take only one paragraph in the whole manuscript.
- The authors appear to have chosen the only parameter – SL. However, nothing is told about combining the complex metric based on a set of parameters.
- Table 1 appears misguiding since it does not perform any grouping or clarifying: the same parameters, namely, ‘stride speed’ and ‘stride length’, present twice. Perhaps it may be presented in the matrix form where the columns designate parameters and the rows do system; so the cells could contain the mark (like a tick) to show that that parameter measurement was possible by means of that system.
- Matrix Q0 at line 229 would be looking much better if it was written as a common matrix, not as a string expression with commas and semicolons.
- Equation 2 contains an excessive brace.
Abstract
Abstract provides concise summary of the paper and its main results.
Language
There are almost no mistakes (the reviewer has found one at line 142 ‘Its uses’) but the sentences are almost completely lack of linking or signposting phrases. This makes the whole text a bit uncommon and more difficult to read.
Conclusion
The reviewer considers the paper rather novel and goal-oriented. It may well become interesting to many readers. However, the current version of the paper cannot be recommended for publication unless the noticed drawbacks are corrected or clarified.
Author Response
Dear Reviewer,
We thank you for careful reviews of our manuscript and the comments and suggestions you provided to improve the quality of the manuscript. You have brought up good remarks and we appreciate the opportunity to clarify our research objectives and results. The following responses have been prepared to address all of your comments in a point-by-point fashion in order to clarify several issues you raised. Please find attached our revised manuscript and below a summary of how we responded to the comments (in blue). The modifications have been highlighted in the revised paper.

Reviewer 2 Report
he paper “Risk of Falling in a Timed Up and Go test Using an UWB Radar and an Instrumented Insole” presents a combined system that is based on UWB radar and instrumented insole for TUG test monitoring. However, many issues need to be addressed which are shown as follows.
Comments:
Point 1: The authors used the proposed risk of falling analysis system as proof of concepts by validating the results with only one participant. However, larger sets of participants are still required to execute experiments because any paraments set in this work could be different from the increasing number of participants. Moreover, the structure and methods of this system may show different results caused by different people, especially for PD patients. The main concept of this work is reasonable, but more results should be obtained and analyzed to verify the robustness of the proposed system.
Point 2: In figure 10, the stride length estimation of UWB radar seems distributed more than the other three approaches, which means that the UWB radar positioning results have the lowest precision. However, the stride length estimation p-values of the approaches are compared with UWB radar positioning results in Table 2. Thus, the UWB radar positioning accuracy need to be improved so the conclusion of comparing approaches’ performance will be reasonable.
Point 3: Three approaches that used the y-axis acceleration to compute the stride length are shown in section 2.5.1. This system will acquire the gait data first and then use the data to compute the risk of falling score in back-end. Thus, the system has no requirements to consider the time complexity of algorithm for calculating the gait paraments. Nevertheless, the approaches were proposed for more than a decade and were not accurate compared with the new methods recently. Please state the reason why choose these approaches to estimate stride length.
Point 4: The data processing of UWB radar signal is based on the KF which can reduce the estimation error and noises. However, this method is already widely used in UWB-based positioning system with higher sample rate. In this system, the KF updates at the 5 Hz which is much lower than the others’ application. Please state the influence of the low sample rate and how this situation affect the stride length accuracy.
Point 5: In section 2.1, the abbreviation of CE and FCC need to be defined as the full name first.
Point 6: The threshold value of 0.4 on the lines 291 for PD persons, particularly, needs to be better explained because the threshold is related to assess the PD gait paraments which will influence the risk of falling score.
Point 7: Equation (1) mentioned in the section 2.4.2 computes two values, PL and PR. However, the values aren’t used in the following article. Please state the values meaning and how to use them.
Point 8: In equation (6), the definition of Sk needs to be clearly explained because the equation shows this same algebra twice, but they have different meanings.
Point 9: In equation (7), the algebra of DIns,i and DRad,i are not equal to the following sentence (line 408) which is stated the meaning.
Point 10: In the section 1 (Introduction), the authors mentioned fall detection and monitoring solutions. I suggest adding the paper “Innovative Head-Mounted System Based on Inertial Sensors and Magnetometer for Detecting Falling Movements” which is published on MDPI Sensors Journal in the reference list.
Author Response
Dear Reviewer,
We thank you for careful reviews of our manuscript and the comments and suggestions you provided to improve the quality of the manuscript. You have brought up good remarks and we appreciate the opportunity to clarify our research objectives and results. The following responses have been prepared to address all of your comments in a point-by-point fashion in order to clarify several issues you raised. Please find attached our revised manuscript and below our responses (in blue) to your comments follow. The modifications have been highlighted in the revised paper.

Round 2
Reviewer 1 Report
The authors accurately replied to the comments and remarks in the review I gave at previous round. All comments were explained and the necessary correction was made to the paper.
Nevertheless, one point requires additional attention. The authors gave the reply for my second comment, for each of its points. The reply presents the accurate and exhaustive explanation for a key point of the paper. However, for some reason, this reply (or its adaptation) has not been properly incorporated in the enhanced version of the paper:
-equation (6) still looks corrupt or made of eastern characters;
- the entire reply for comment 2b is not included but highly recommended for adding to the paper (with changing the corresponding lines of Section 2.5.2);
- the formatting of this text in the reply looks better than what I can see in the paper.
- the value of Sk cannot be greater than 100, since Rk is non-negative according to its definition.
Conclusion
Equation 6 must be corrected. The paper can be recommended for publication, but the above-mentioned minor changes are highly appreciated.
Author Response
Dear Reviewer
We thank you again for careful reviews of our manuscript and the comments you provided to improve the quality of this manuscript. Please find attached our revised manuscript and below a summary of how we responded to your comments. The modifications have been highlighted in the revised paper.
